# The Interplay Between Juvenile Delinquency and ADHD: A Systematic Review of Social, Psychological, and Educational Aspects

**DOI:** 10.3390/bs15081044

**Published:** 2025-08-01

**Authors:** Márta Miklósi, Karolina Eszter Kovács

**Affiliations:** 1Institute of Education and Cultural Sciences, Faculty of Humanities, University of Debrecen, Egyetem tér 1, 4032 Debrecen, Hungary; miklosimarta@unideb.hu; 2Department of Counselling, Developmental and School Psychology, Institute of Psychology, Faculty of Humanities, University of Debrecen, Egyetem tér 1, 4032 Debrecen, Hungary

**Keywords:** ADHD, juvenile delinquency, comorbidity, risk factors, early intervention

## Abstract

Attention deficit/hyperactivity disorder (ADHD) is a neurodevelopmental disorder characterised by inattention, hyperactivity, and impulsivity, frequently observed in juvenile offenders. This systematic review explores the interplay between ADHD and juvenile delinquency, focusing on behavioural, psychological, and social dimensions. Following the PRISMA guidelines, a systematic literature review was conducted using EBSCO Discovery Service, Science Direct, PubMed, and snowballing techniques. Studies meeting specific inclusion criteria, including juvenile offenders diagnosed with ADHD and comparisons to non-offender or non-ADHD control groups, were analysed. The methodological quality of studies was assessed using the Joanna Briggs Institute appraisal tools. A total of 21 studies were included, highlighting significant associations between ADHD and juvenile delinquency. ADHD symptoms, especially impulsivity and emotional dysregulation, were linked to an earlier onset of offending and higher rates of property crimes. Comorbidities such as conduct disorder, substance use disorder, and depression exacerbated these behaviours. Sociodemographic factors like low education levels and adverse family environments were also critical modifiers. Early intervention and tailored treatment approaches were emphasised to address these challenges. The findings underscore the need for early diagnosis, individualised treatment, and integrative rehabilitation programmes within the juvenile justice system to mitigate long-term risks and promote social inclusion.

## 1. Introduction

Attention deficit/hyperactivity disorder (ADHD) is a mental condition linked to nervous system dysfunction, primarily characterised by hyperactivity, impulsivity, and attention deficit ([90]). ADHD can be attributed to both genetic and environmental influences. Studies examining hereditary factors indicate that first-degree relatives face a 3–5-times-higher risk of developing ADHD, while twins show a risk ranging from 65 to 90% ([33]). In the early 1900s, the initial understanding of ADHD suggested it stemmed from disruptions in moral controls and related motivational issues that were often apparent to those around them ([64]). Thanks to Faraone’s research, it has become increasingly clear that ADHD has a clear molecular biological and genetic background in addition to environmental risks ([33]). A study conducted by [26] ([26]) demonstrated that ADHD has an average heritability rate of 76%, ranking it among the most genetically inherited psychiatric disorders. Additionally, factors such as complications during birth, maternal smoking, and familial issues may also play a role ([30]; [91]). Adverse childhood experiences (ACEs) may also be relevant factors, as ACEs are common among juvenile offenders. [58] ([58]) found that youth involved in the juvenile justice system were more than 12 times more likely to have experienced at least one ACE than their uninvolved peers. Research confirmed that they may influence the development of ADHD and the onset and maintenance of delinquency ([29]).

Childhood ADHD prevalence estimates range between 4 and 12% ([27]), and follow-up research indicates that 70–80% of those diagnosed continue to exhibit symptoms into adolescence ([30]; [91]). The prevalence rate is two to four times greater in males than in females ([9]; [12]; [14]). These differences in gender are noticeable at an early age but tend to diminish as individuals grow older. Clinical studies reveal that boys display a higher incidence of hyperactivity–impulsivity than girls ([9]; [12]). Additionally, meta-analyses show ADHD prevalence rates of 26–30% among juvenile and adult detention populations, reflecting a risk that is five to ten times greater than that of the general population ([7]).

Disruptive behaviour patterns are often exhibited by children with elevated levels of ADHD, especially during their teenage years, indicating that hyperactivity might serve as a predictor for antisocial behaviour ([6]). While numerous studies have shown that children diagnosed with ADHD face a heightened risk of delinquency, the clarity surrounding ADHD’s role in predicting delinquency in both clinical and population-based studies remains inadequate ([9]; [62]). Barkley’s research revealed that children with ADHD were more inclined to partake in antisocial activities than their peers and emphasised that the frequency of these atypical behaviours was a strong indicator of ADHD severity during childhood and adolescence ([10]).

A study in Finland found that more than 50% of prison inmates met the diagnostic criteria for ADHD, compared to 45% in Germany ([50]; [86]). In a Korean study evaluating 98 juvenile detainees and 84 non-offending controls, 42.4% of adolescents with a history of delinquent offending were diagnosed with ADHD, compared to only 11.9% of the control group ([19]).

While the research results are clear, the mechanisms that might connect ADHD with delinquency remain largely unexplored ([98]). Caution is warranted when considering these connections, as hyperactivity and attention deficit frequently co-occur with other early risk factors, including conduct disorder, familial disadvantage, and low verbal intelligence ([91]). Research has established the link between ADHD and juvenile delinquency, with studies showing that children and adolescents diagnosed with ADHD are at higher risk of delinquency than those without ADHD, particularly in the presence of comorbid conduct disorder ([38]; [94]). The core symptoms of ADHD (impulsivity, inattention, and hyperactivity) can contribute to poor decision making and difficulty complying with rules, which increases the propensity to delinquent acts ([38]; [39]; [94]). Hence, the mechanisms of the link between ADHD and delinquency are likely to include a lack of reasoned decision making, impulsivity, inattention, higher risk taking and hyperactivity, and a higher risk factor for committing violent acts. These conditions contribute to poor decision making, rule-breaking, emotional dysregulation, and executive dysfunction ([85]). Different ADHD symptom profiles can lead to different patterns of offending; for example, impulsive ADHD is associated with impulsive crimes such as robbery, while inattentive ADHD is associated with more premeditated crimes ([85]). Additionally, the influence of ADHD on delinquency may be shaped by associated negative factors, such as educational failure, which is commonly observed in the lives of young individuals affected by this condition ([86]; [98]).

Research by Moffitt emphasises that the transition from youth to adulthood represents a crucial phase concerning the age crime curve. He differentiates between life-course-persistent offenders and adolescence-limited offenders in his analysis of severity and continuity, noting that juvenile delinquency is typically the norm rather than an anomaly ([69]). According to Moffitt’s dual taxonomy, the distinction between life-course-persistent offenders and adolescent-restricted offenders lies in the early emergence of antisocial behaviour ([70]). The main contributors to adult delinquency are neuropsychological deficits, which may be inherited or developed during early childhood. It is suggested that growing up in a challenging family setting heightens the criminogenic potential of these risk factors ([69]). Consequently, ADHD is viewed as a reflection of a neuropsychological deficit that could lead to a chronic and persistent trajectory toward delinquency ([34]). [68] ([68]) proposed that early risk factors directly influence delinquency in adulthood rather than being influenced by social factors during later adolescence. While the theory acknowledges that hyperactivity and impulsivity can lead to adverse outcomes throughout life, it views these results as downstream effects—distinct expressions of a shared syndrome—that contribute minimally to the overall causal sequence ([91]).

[89] ([89]) proposed that the majority of antisocial children do not transition into criminality, asserting that an antisocial upbringing is not a prerequisite for delinquency in adulthood. Adhering to the fundamental principle of social control theory, they contend that weak connections to prosocial figures lead to delinquent behaviour ([89]). Consequently, [34] ([34]) posits that while ADHD is not a definitive cause of adult delinquency, it serves as a risk factor that may influence delinquent behaviour by obstructing robust ties to conventional institutions.

During the teenage years, the primary sources of social control are attachments to parents, peers, and school, and issues related to hyperactivity and inattention can significantly impact these connections. Furthermore, research indicates that children diagnosed with ADHD are less frequently raised by both biological parents ([38]) and that their parents often face challenges such as low socioeconomic status and criminal histories ([35]). Such traits are typically linked to diminished levels of social control ([30]). Additionally, irrespective of the characteristics of the parents, the tendency of ADHD to manifest as poor interpersonal skills likely affects parenting styles and socialisation practices ([93]). Research has shown that inadequate parental discipline, low-quality parent–child relationships, and weakened family cohesion exacerbate the connection between ADHD and antisocial behaviours ([98]).

Based on the literature review, we can state that juvenile offenders are a very complex population, and it becomes important to have a global vision of the adolescent, including family and individual risk factors. This systematic review aims to shed light on the interplay between juvenile delinquency and ADHD through exploring behavioural, social, and psychological aspects. The review considers some traits associated with the disorder, such as emotional dysregulation and impulsivity, but neglects the early onset of criminal behaviour as a fundamental aspect. Furthermore, we make an attempt to analyse the additional factors such as conduct disorder, substance abuse, and depression that accompany the disorder and increase the propensity to delinquency. Several family or sociodemographic variables, including education, employment status, and parental support, are also believed to be of great importance in modifying the relationship between ADHD and delinquency. Also, the treatment needs of the patients are highly dependent on the mechanisms hypothesised to mediate the relationship between ADHD and criminality, thus calling for individualised and holistic treatment approaches.

## 2. Materials and Methods

This systematic literature review was created based on the Preferred Reporting Items for Systematic Reviews and Meta-Analyses (PRISMA) guidelines ([71], see Figure 1).

### 2.1. Literature Review

The EBSCO Discovery Service Search Engine, Science Direct, and PubMed were used for systematic search. Using the Boolean search string, the following keywords we applied for searching were as follows:
(“juvenile delinquency” OR “juvenile offenders” OR “youth offenders” OR “juvenile justice” OR “juvenile corrections”)

AND
(“ADHD” OR “attention deficit hyperactivity disorder” OR “attention deficit-hyperactivity disorder”).

Beside the search engine, snowball search was applied, checking the reference list of papers found by the search engine. Also, the top 30 journals indexed in Web of Science (having the highest percentile above 50%) were screened (see Appendix A, Table A1). Journals were selected based on their indexing in Scopus and Web of Science and their impact metrics, including citation percentile and relevance to the fields of psychology, psychiatry, education, and criminology. The searches were performed in August 2024. Unscreened articles were listed in Zotero (V6.0.22, Roy Rosenzweig Center for History and New Media, George Mason University, Washington DC, USA).

### 2.2. Inclusion and Exclusion Criteria

The following inclusion criteria were set, following the PICOS format (P: population; I: intervention; C: comparison; O: outcome; S: study design):*Population*: Juvenile offenders/criminals;*Intervention*: Original empirical research published in a peer-reviewed journal;*Comparison*: Examined juvenile offenders diagnosed with ADHD compared to those without ADHD diagnosis or with comorbid issues in various contexts (sociodemographic background, nation, psychological characteristics, or non-offenders as a control group);*Outcome*: Behavioural outcomes and criminal offending and academic achievement;*Study design*: Observational, interview, survey, cohort study, or randomised controlled trial.

Papers had to be written in English, published between 2004 and 2024, and in the disciplines of psychology, social sciences, humanities, and educational sciences. Language restrictions have been applied to ensure the consistent application of quality appraisal tools and to minimise the risk of misinterpretation due to translation inaccuracies. Review papers, commentaries, letters to the editor, conference papers, books, book chapters, dissertations, or newspaper articles were excluded. Grey literature (e.g., dissertations, conference proceedings, and government reports) have also been excluded to maintain a focus on peer-reviewed, methodologically rigorous studies.

### 2.3. Data Extraction and Assessment of Methodological Quality

A comprehensive multistage screening procedure was implemented to identify studies meeting the inclusion criteria. The authors independently searched the literature and examined the titles and abstracts of every study. Following this, all identified records were subjected to a screening of their titles and abstracts. Studies that met the inclusion criteria then proceeded to an exhaustive full-text evaluation. The authors oversaw the meticulous analysis, quality assessment, and data extraction of the chosen studies. In cases of ambiguity, discussions among the authors were held to arrive at a consensus.

An Excel spreadsheet along with data extraction forms were used for data extraction. We included the full article citation, sample characteristics (number of participants, gender, ethnicity, diagnosis of ADHD, comorbid disorders), aim of the paper, methods (qualitative or quantitative), tools applied, results/outcome, and comments related to study quality. Most studies carried out cross-sectional research regarding the methodological framework. Some studies conducted long-term follow-ups, up to 15 years, to examine recidivism and crime patterns. Some studies combined data from interviews, questionnaires, and official criminal records. Therefore, the risk of bias and quality of the studies was evaluated using the Joanna Briggs Institute (JBI) critical appraisal tool, focusing on cross-sectional studies ([73]), qualitative studies ([60]), case series ([77]), and randomised controlled trials ([8]). Each article was assessed with the appropriate tool on a 4-point scale (yes/no/unclear/not applicable).

## 3. Results

Overall, 989 records were detected. After double filtering, 121 records were excluded, and after title and abstract screening, 823 records were excluded. During the title and abstract screening phase, the majority of the 823 excluded records were omitted because they (a) did not focus on juvenile offenders, (b) did not address ADHD explicitly, (c) were review articles or grey literature, or (d) lacked empirical data relevant to our outcomes of interest. Therefore, 45 papers were sent for full-text screening, which led to the involvement of 21 papers in the qualitative synthesis.

The studies have been conducted in various countries, including Germany ([11]; [47]; [54]; [79]; [82]), the United States of America ([56]; [94]; [95]; [100]), Hong Kong ([80]), Russia ([59]), South Korea ([21]), the Netherlands ([88]), India ([43]), Sweden ([97]), and Nigeria ([3]).

### 3.1. Sociodemographic Factors

The study’s measurement of the relevance of sociodemographic factors was under-represented. Regarding gender, most studies measured only male offenders ([3]; [43]; [47]; [54]; [56]; [59]; [79]; [80]; [82]; [88]; [94]). Studies investigating only female offenders did not appear. Focusing on both sexes without any comparisons was typical in two cases ([90]; [97]), while the investigation of gender differences was unclear in three studies ([21]; [49]; [63]). Gender comparison appeared in two studies ([11]; [95]; [100]). In the study of [11] ([11]), male participants were over-represented in the low ADHD subtype, while females were over-represented in the severe ADHD subtype. They also reported that young women who committed offences tended to exhibit more severe ADHD, with or without intermittent explosive disorder (IED), along with additional behavioural issues; however, there were no significant differences noted in terms of cumulative adverse childhood experience (ACE) burden. [95] ([95]) reported no significant difference in the average age of initial incarceration for both boys and girls. When compared to the control group, those with ADHD, regardless of gender, were twice as likely to engage in such offences. While boys and girls exhibited a comparable pattern of first offences, the data for girls were limited. Children diagnosed with ADHD, both boys and girls, are notably more prone to having records of community corrections and incarceration compared to those without the disorder. While boys with ADHD were more inclined to have a community correction record at an earlier age, this trend was not observed in girls. However, the highest number of offences occurred among individuals aged 15 to 17.

In addition, only a few studies focused on social status and family-related factors. The study of [100] ([100]) exclusively focused on the severity of violent behaviour in light of sociodemographic factors and ADHD. The results stated that individuals who qualified for an ADHD diagnosis at the baseline were most likely to be placed in the High Chronic violent offending category. The probability of being categorised into the Desisting and Moderate Stable trajectory groups, as opposed to the Abstaining group, was notably increased for individuals who met the ADHD diagnostic criteria at baseline. Additionally, having an ADHD diagnosis at baseline not only predicted involvement in violent offences but also indicated a potential increase in the frequency of offences committed by juvenile offenders. When it comes to gender, males exhibited a significantly higher likelihood of being assigned to the Moderate Stable, Desisting, and High Chronic groups compared to the Abstaining group. In relation to race, only the assignment to the Moderate Stable trajectory was significantly affected, with Black participants facing a considerably greater relative risk of being placed in this trajectory group than their White counterparts. Socioeconomic status (SES) did not significantly influence the relative risk of assignment to any of the trajectory groups.

[88] ([88]) revealed disparities among adolescents concerning comorbidity, education, and living conditions. Comparing juvenile offenders diagnosed with ADHD and/or ASD, the group diagnosed solely with ADHD demonstrated a lower level of completed education and had fewer adolescents living with both biological parents. [82] ([82]) stated that individuals with ADHD were notably younger, had lower educational attainment, and experienced higher unemployment rates when compared to those without ADHD. Additionally, the age at which they first faced conviction was younger, and the incidence of delinquent behaviour before reaching the age of criminal discretion was greater than that observed in young prisoners who did not have ADHD.

[97] ([97]) investigated how psychosocial background elements, such as age at first conviction and substance abuse or dependence on primary relatives, along with clinical factors like ADHD diagnosis, IQ score, and the age at which drug abuse began, influence the persistence of violent criminal behaviour. Their findings indicated that age at first conviction plays a significant role in persistent violent criminality, suggesting that an earlier first conviction correlates with an increased risk of developing ongoing violent behaviour.

The study of [11] ([11]) highlighted that many young offenders carry the weight of adverse childhood experiences. Over 85% of those surveyed reported encountering at least one of the five evaluated ACEs, while more than 28% stated they had faced at least four out of the five categories. High ADHD severity was related to increased ACE rates; the cumulative ACE score predicted severe ADHD, but only when coexistent with IED. Therefore, elevated levels of ACEs are found not only among intensive and chronic young offenders but also among those delinquents who are in the early phases of their criminal development.

### 3.2. Type of Crime

Offences can be grouped according to several criteria, one of which is the distinction according to the object of the offence. [84] ([84]) points out that the police more often arrest juveniles for crimes against property than for crimes against a person. In particular, according to researchers, “property crime, carrying a concealed weapon, illegal drug possession, and arrest rates have been shown to be positively related to ADHD status” ([14]; [37]; [62]).

The predominance of crimes against property has been highlighted in some research ([11]; [47]; [59]; [79]; [88]; [95]). [79] ([79]) found that boys with ADHD were more likely to be involved in non-violent property crimes (property crime = 58.6%) than crimes against the person (assault = 21.8%). Similar results were found by [11] ([11]) (crimes against property = 35.9%; assault = 21.8%) and [47] ([47]), although the difference was smaller (crime against property = 34.8%; assault = 33.3%). [88] ([88]) found a significant correlation in their study, showing that the proportion of non-violent crimes against property was significantly higher among people with ADHD (22, 24%) than among people with ASD. When disaggregating the crimes, it was found that the most common crimes among the group of adolescents with ADHD were violent property crimes (37.8%), non-violent property crimes (24.4%), and moderate violent crimes (21.1%). [59] ([59]) also found in their study that most juveniles were convicted of crimes against property (theft, car theft, etc.; 51%), followed by crimes related to violence (e.g., assault, robbery; 38%), and the smallest proportion of juveniles in the sample were convicted of sexual violence (6%) or homicide (5%). In their study, [95] ([95]) categorised offences into eight groups, but of these, burglary was also the most common, with this offence being twice as likely in boys with ADHD as in boys without ADHD.

A study conducted by [63] ([63]) revealed contrasting findings, indicating that juvenile offenders responsible for crimes against individuals exhibited a higher prevalence of ADHD symptoms (18%) and behavioural issues (20%) compared to those involved in property crimes as well as alcohol- and drug-related offences. Similar results were found in research by [54] ([54]), comparing instrumental violent crime (violence used as a means to achieve an external goal, e.g., robbery) and reactive violent crime (motivated by anger, revenge, or frustration, in which the goal of the violent act is primarily to harm the victim). His research found that the juveniles studied were more likely to have committed reactive violent crime (*n* = 50) than instrumental violent crime (*n* = 39).

In their research, [3] ([3]) highlighted a particularly intriguing outcome regarding the types of crime in Africa, revealing that over two-thirds of the juveniles analysed were apprehended for status offences—actions deemed non-criminal that are classified as violations solely due to the offender’s minor status. These offences encompassed truancy, running away from home, vagrancy, violations of curfew, associating with dangerous adults, and consistently disregarding household rules. This finding aligns with comparable studies in Nigeria ([4]; [53]; [78]) across Africa ([76]).

### 3.3. Age of Onset

Of the studies reviewed, five examined the relationship between age at first offence and involvement with ADHD. The studies found similar results, suggesting that there is consistent evidence that individuals with ADHD start offending at a younger age.

[82] ([82]) found that ADHD offenders have a lower average age at first arrest and a higher rate of offending at age 14 compared to juveniles who do not have ADHD. In their study, [97] ([97]) conducted an ROC (receiver operating characteristics) analysis to evaluate the continuous variable of age at first conviction in relation to the persistence of violent criminality, aiming to enhance its predictive accuracy. This relatively straightforward variable significantly predicted the continuation of violent criminal behaviour, achieving an AUC of 0.69 (AUC = area under the ROC curves).

A study conducted by [94] ([94]) revealed that boys diagnosed with childhood ADHD alone and those with both childhood ADHD and ODD faced a heightened risk of delinquency by age 18. These two groups were notably more inclined to have committed prior offences, recorded more offences, and engaged in more severe offences than their juvenile counterparts. The implications of these findings indicate that boys with ADHD who displayed minimal antisocial behaviour during primary school still face the potential for future criminal activity. This concern is significant, as early delinquency and engagement in various crimes correlate with an atypical delinquency trajectory ([67]; [69]). Additionally, this outcome aligns with other research indicating that adolescents with ADHD tend to begin committing crimes at an earlier age than those without the condition ([39]; [69]). Regarding serious delinquency, both boys with ADHD only (23.4%, OR = 1.84) and those with ADHD + ODD (25.4%, OR = 2.01) exhibited comparable levels of risk.

According to [95] ([95]), males diagnosed with ADHD engaged with the justice system for the first time at an earlier age than those in the control group. Typically, the average age for their initial involvement with community correction or incarceration ranged from 15 to 17 years. Among the children with ADHD who appeared in the community correction register, 308 boys (39%) were aged between 10 and 14 years, whereas the control group had 235 boys (29%) in the same age range. Boys with ADHD received their first community correction record at a younger average age compared to their counterparts in the control group, with ages of 15 years and 9 months versus 16 years and 3 months.

[28] ([28]) examined differences in the onset of ADHD between young people with and without the disorder. Youths with ADHD showed significantly earlier antisocial onset than youths without ADHD. Children with ADHD broke the rules for the first time 1.1 years earlier, had their first contact with the police 1.3 years earlier, and had their first juvenile court appearance 0.8 years earlier.

### 3.4. Psychological Consequences

Some studies have used a diagnostic criteria system (DSM-IV, DSM-5, ICD-10) to identify ADHD, ODD, CD, or other comorbid disorders ([3]; [23]; [63]; [88]; [90]; [94]; [95]; [97]). In addition, diagnostic interviews (e.g., K-SADS) have been used in some cases ([56]; [59]). However, several studies have used self-completion questionnaires to map behavioural and psychological characteristics, even for ADHD categorisation ([11]; [21]; [43]; [47]; [79]; [80]; [82]). In the case of some studies, the diagnostic criteria for assigning offenders to the ADHD group were unclear, making it difficult to determine how participants were classified ([3]; [54]; [100]).

[11] ([11]) reported that symptoms of ADHD were notably common, with one-fourth of the participants indicating at least moderate symptoms and another quarter experiencing severe symptomatology. The ratio of participants with ADHD was high in the research of [21] ([21]) too; subjects with ADHD constituted 32.4% of the total participants.

Barra et al. mentioned that increased rates of additional internalising and externalising issues were associated with the high severity of ADHD, emphasising the range of psychiatric disorders that often accompany it. Therefore, when considering further psychiatric conditions, it is crucial to take into account not just ADHD but also the presence of coexisting IED symptoms. The results of [82] ([82]) also support these findings. In their research, offenders belonging to the ADHD group exhibited significantly higher scores for internalising issues, such as anxiety, depression, social withdrawal, and somatic complaints. Additionally, the personality dimension of “neuroticism” (measured by the Five-Factor Inventory) showed higher scores, indicating that these subjects were more anxious, depressed, and vulnerable while also displaying increased levels of hostility, self-consciousness, and impulsivity compared to the control group, reinforcing the same conclusion. This was also supported by the results of [43] ([43]), who found a significant difference between juvenile offenders with and without ADHD concerning the severity of their conduct issues. The level of conduct issues was also influenced by the existence of oppositional defiant disorder (ODD). [63] ([63]) also focused on the evaluations of emotional and behavioural issues, which indicated that juvenile offenders experienced both externalising (31%) and internalising (23%) challenges. Internalising issues were primarily characterised by withdrawn or depressed symptoms (13%), followed by anxiety/depression (12%), somatic problems (10%), and affective issues (9%). On the other hand, externalising problems showed huge variation concerning the diagnosis, and they were predominantly represented by oppositional defiant behaviours (19%), followed by rule-breaking actions (13%), conduct issues (11%), and ADHD (10%). These findings align with previous studies that noted a greater prevalence of externalising behaviours, such as conduct disorder, ADHD, and substance abuse, among adolescent offenders compared to typical internalising mental health concerns like depression, panic disorder, and anxiety. This also had a significant impact on close relationships. Children with ADHD experienced a higher rate of peer rejection. They reported fewer close friendships in comparison to those without ADHD, indicating that the long-term impact of ADHD on social interactions is particularly significant for youngsters who continue to have ADHD or exhibit conduct disorder during their teenage years.

The results of the studies focusing on recidivism were ambivalent. [47] ([47]) found that the occurrence of ADHD among prisoners was significantly greater than the prevalence estimates obtained from community samples. Over 50% of the inmates fulfilled the criteria for having ADHD during their childhood. However, in summary, their study’s findings do not confirm the hypothesis that ADHD can forecast criminal recidivism. The analyses conducted did not indicate that ADHD had any effect on survival curves or the likelihood of reoffending. However, the results of [79] ([79]) discovered the significant role of ADHD as a risk factor for recidivism since, based on the results of the 15-year-long follow-up study, offenders diagnosed with ADHD had a significantly faster recidivism rate (2.5 times less time to the next reconviction compared to the offenders without ADHD). The number of further offences and re-incarcerations was also higher in the ADHD subgroup. This study also revealed that ADHD moderates both the relapse and the course of delinquency concerning the number of further engagements in the legal system.

### 3.5. Comorbid Disorders

Depression also appeared as a comorbid disorder. In the study of [21] ([21]), the group experiencing depression was significant, representing 52% of all subjects. Symptoms of ADHD demonstrate a significant positive correlation. A noteworthy negative relationship was found between self-esteem and depression, indicating that lower self-esteem is associated with higher levels of depression. Perceived health state, ADHD symptoms, and self-esteem were assessed as independent variables, while depression served as the dependent variable. Together, these variables explained 37% of the variance in depression, with self-esteem identified as the most significant predictor. The relevance of depression was also measured by [23] ([23]), who stated that conduct disorder was not influenced by depression; however, it indicated that delinquent behaviour or conduct disorder might indirectly impact depression through factors like psychosocial impairment. A relevant correlation was identified between juvenile delinquency and depression and between self-esteem and juvenile delinquency. ADHD was shown as a contributing factor to delinquency, alongside depression, suggesting that the risk factors associated with juvenile delinquency have evolved over time. Compared to the control group, juvenile offenders showed significantly more ADHD symptoms, depression, anxiety, and suicidal ideation.

Obsessive–compulsive disorders are very often detected as comorbid disorders along with ADHD. The findings of [94] ([94]) indicate that all children diagnosed with ADHD face an elevated risk for delinquency, irrespective of comorbidity. When compared to juveniles without ADHD, those with ADHD only, and those with ADHD and ODD, those with ADHD and CD exhibited a greater risk for every index of delinquency, including severity, variety, and age of initiation. The prevalence of severe offending was higher among children with ADHD only and those diagnosed with ADHD and ODD, who also faced a greater risk of starting mild and moderate delinquency at an earlier age, committing a wider range of acts compared to the comparison group. The risks demonstrated by the ADHD-only and ADHD and ODD groups were quite alike, differing only in the variety of offences committed.

Conduct disorder is also a frequently diagnosed comorbid disorder. In the study of [56] ([56]), the group with CD/ADHD received higher ratings on individual factors of the Structured Assessment of Violence Risk in Youth (SAVRY) due to the inclusion of items such as ‘risk-taking/impulsivity’ and ‘attention deficit/hyperactivity difficulties.’ Additionally, they were rated significantly higher on lifestyle factors of the Psychopathy Checklist (PCL: YV), which features ‘stimulation seeking’ and ‘impulsivity’ as components. Furthermore, the CD/ADHD group scored notably higher on the social factor of the SAVRY, which encompasses aspects like peer rejection, ineffective parental management, stress, inadequate coping skills, and a lack of personal support. [59] ([59]) had similar results, revealing that ADHD is frequently encountered and almost invariably seen alongside CD. This particular diagnostic pairing was linked to elevated levels of comorbidity, more complex substance abuse issues, disruptive behaviours, and a significant prevalence of PTSD. The significant occurrence of PTSD linked to ADHD/CD carries a crucial clinical implication by emphasising the heightened risk of trauma exposure within this population. The current research indicates that individuals with a dual diagnosis of ADHD and CD exhibit greater levels of disturbance, particularly in terms of psychiatric comorbidities and heightened aggressive and disruptive behaviours.

Autism spectrum disorder (ASD) also appeared as a comorbid disorder in the studies analysed. [88] ([88]) investigated juvenile offenders diagnosed primarily with ADHD and ASD, among whom 36.7% had ASD, 47.9% ADHD and 15.4% both ASD and ADHD. Besides these two disorders, disruptive behaviour disorder (36.2%), substance disorder (20.2%), and reactive attachment disorder (5.3%) appeared as comorbid disorders. In their study, no significant difference was found in the distribution of violent offences across diagnostic groups; however, individuals diagnosed solely with ASD exhibited a significantly higher likelihood of sex offending compared to those with ADHD, whether diagnosed alone or alongside ASD. In contrast, adolescents with ADHD faced a greater propensity for non-violent property offences than their ASD counterparts, whether diagnosed alone or in combination.

Intermittent explosive disorder was also investigated in one study. In the research of [11] ([11]), a significant occurrence of DSM-5-oriented IED (36%) was observed when compared to rates documented in the general population, psychiatric patients, or other offender groups. This result suggests that IED is prevalent among young offenders and deserves further scientific attention.

The relevance of psychopathic personality traits must also be mentioned. [28] ([28]), focusing on institutionalised delinquents, stated that rule-breaking behaviours appear early. The only significant predictor of total arrests and self-reported delinquency was the initiation of police contact or arrest. Outcomes were not predicted by either the onset of rule-breaking or the referral to juvenile court. In terms of psychopathic personality traits, only the onset of rule-breaking correlated with the level of psychopathy. Youths diagnosed with ADHD demonstrated a notably earlier onset of antisocial behaviour compared to those without the condition. Children with ADHD began violating rules 1.1 years ago. Youths with conduct disorder experienced an arrest significantly earlier than those without the disorder, having first been contacted by police 1.3 years prior and referred to juvenile court 0.8 years earlier. Children and adolescents with CD exhibited an earlier average onset across all three types, occurring 0.9 years sooner than those without CD. In another study, [54] ([54]) stated a weak relationship between psychopathy and ADHD. Indicators of ADHD had a negligible impact on the prediction of violent behaviour compared to the evaluation of juvenile psychopathy. ADHD itself did not increase the likelihood of violent institutional behaviour; however, along with the higher level of psychopathy, the chance of violent behaviour increased. In the research of [56] ([56]), the Psychopathy Checklist’s interpersonal factor revealed significantly higher scores for the CD/ADHD group; this factor embodies the fundamental personality characteristics associated with psychopathy, such as grandiosity, manipulation, and pathological lying.

Substance use disorder (SUD) may also be a comorbid disorder among juvenile offenders. The follow-up study of [97] ([97]) aimed to examine institutionalised adolescents, categorising them into three groups: those with comorbid SUD and ADHD, those with SUD without ADHD, and those without SUD. Conducted over an average span of approximately three years post-institutionalisation, the study sought to assess rates of criminal behaviour, utilisation of inpatient healthcare services, and premature mortality and to investigate whether risk factors associated with group classification correlated with ongoing violent criminality. The overall results for these three groups were similar, marked by significant antisocial behaviours resulting in new sentences for many individuals, alongside a notable prevalence of illness. The SUD plus ADHD group reported a notably higher total of criminal acts charged in all convictions, with their numbers being twice as high as those observed in the non-SUD group. In this group, the inclination towards associating with delinquent peers for the planning and execution of crimes was less pronounced. Thus, their criminal activities relied less on being part of a similar peer group, as they predominantly committed offences independently. This suggests that individuals within the SUD plus ADHD group are more closely associated with an earlier initiation and a broader antisocial lifestyle, frequently engaging in criminal acts on their own rather than as members of a group of delinquent youths.

As we can see, several comorbid disorders may appear along with ADHD. However, it was not typical in the studies to discover a wide range of comorbid disorders and their potential impact on the behaviour of juvenile offenders. [90] ([90]) investigated the relevance of conduct problems (ODD/CD symptoms) and substance use as mediators of risk concerning risky sexual behaviour and ADHD. They stated that mediation models indicated a direct link at first between symptoms of ADHD and self-reported sexual risk behaviour. This effect was explained by the separate pathways of problematic use of alcohol and marijuana, while conduct problems did not account for it. The connections between ADHD, substance use issues, and sexual risk behaviour (RSB) varied based on the existence of comorbid conduct problems in youths. In particular, the influence of ADHD on sexual risk behaviour was limited to a specific group of youths displaying notable comorbid conduct issues and was entirely mediated by problematic marijuana use. In contrast, there was no direct or indirect relationship between ADHD and RSB in youths who did not have heightened conduct problems. Overall, these findings suggest that the link between ADHD and RSB is indicative of the degree to which comorbid conduct issues, and especially problems related to substance use, have emerged.

Finally, the expenses of the offenders were measured only in the study of [49] ([49]), who found that, on average, service expenses for the ADHD group surpassed those of the non-ADHD group by USD 25,000 per child. For young individuals identified as having both ADHD and CD, the average costs for services reached over USD 80,000 across six years—more than double the expenditure for those with only ADHD and six times higher than that of a child without either disorder. Expenditures differ not just in their level but also in their composition. Further analysis of the data revealed that, when averaged over the years, there is a positive correlation between school service costs and juvenile justice costs for youth with ADHD, a pattern that does not appear in youth with CD or those with comorbid conditions. These statistics imply that for individuals with an ADHD diagnosis (but not CD), the requirement for support in school may indicate broader disciplinary issues that remain undiagnosed.

### 3.6. Academic Achievement

Of the studies reviewed, five were concerned with the educational attainment of young people. Their findings are in line with previous research suggesting that one of the main consequences of ADHD may be poorer academic outcomes as a result of inattention, overactivity, and impulsivity ([9]; [37]).

The studies by [82] ([82]) and [88] ([88]) showed that the educational attainment of the group of young people diagnosed with ADHD alone was lower than in Rutten’s study for young people with ASD (autism spectrum disorder) or ASD alone and in Retz’s study for young people without ADHD. The research by [63] ([63]) examined educational attainment by crime type, comparing property crimes, alcohol/drug-related crimes (54%), and crimes against people. It found that adolescents who committed property crimes (60%) and alcohol/drug-related crimes (54%) had higher rates of irregularity in their school careers than those who committed crimes against people. [80] ([80]) investigated the reading ability of juvenile detainees in addition to ADHD. It found that juveniles with ADHD were associated with negative academic orientation; their difficulties with academic orientation were reflected in poor grades, less serious academic goals, and poorer planning. Interestingly, this is the only area where worse outcomes were reported for youth with ADHD compared to RD (reading disability); apart from this, it can be concluded that ADHD was associated with fewer psychosocial difficulties than RD.

In the study conducted by [4] ([4]), the School Engagement Measure (SEM; [41]) was utilised to evaluate the engagement levels of respondents in school three months following their enrolment. The assessment covered all three aspects of school engagement: behavioural, emotional, and cognitive. Notably, the presence of a symptom cluster that met the DSM-5 diagnostic criteria correlated with significantly lower mean SEM scores across any of the three behavioural disorders examined among the respondents. Atilola’s research revealed that, on average, approximately two-thirds of the respondents had dropped out of school three years prior to their arrest or detention. Furthermore, there was an average gap of about four years between the anticipated duration of formal education at the time of arrest and detention. A clear link was identified between behavioural issues, particularly the symptom complex associated with ADHD, and increased dropout rates as well as lower educational outcomes for these individuals in comparison to their peers.

### 3.7. Methodology

Some studies used a diagnostic criteria system (DSM-IV, DSM-5, ICD-10) to identify ADHD, ODD (oppositional defiant disorder), CD (conduct disorder), or other comorbid disorders ([3]; [23]; [63]; [88]; [90]; [94]; [95]; [97]). In addition, diagnostic interviews (e.g., K-SADS) have been used in some cases ([56]; [59]). However, several studies have used self-completion questionnaires to map behavioural and psychological characteristics, even for ADHD categorisation ([11]; [21]; [43]; [47]; [79]; [80]; [82]). In the case of some studies, the diagnostic criteria used were unclear, which determined the criteria used to assign offenders to ADHD in the study ([3]; [54]; [100]).

Several researchers ([47]; [95]) caution against establishing a clear link between mental health problems and offending. [95] ([95]) point out that the criminalisation of young people with mental health disorders has been raised as a concern by mental health professionals, advocacy groups, and researchers. [47] ([47]) raise doubts along the recidivism line when they point out that the results of this study highlight the need for a theoretical and practical distinction between risk factors for delinquency and risk factors for criminal recidivism.

The significance of early intervention and crime prevention is underscored by a majority of studies ([3]; [11]; [21]; [43]; [59]; [79]; [86]; [90]; [95]; [100]), which indicate that failing to address adolescent ADHD early on may lead to its progression into adult ADHD or the emergence of antisocial behaviours. More specific recommendations are provided by [11] ([11]), [95] ([95]), and [100] ([100]), who emphasise the essential need for standardised yet individualised assessments of the risks and needs of young offenders or high-risk groups, conducted by professionals with psychiatric or psychological training ([11]). According to [95] ([95]), a comprehensive psychometric and mental health evaluation should be mandated for children entering the justice system, highlighting the necessity for thorough mental health screenings for all youth involved in the juvenile justice system ([100]).

Improvement in training health professionals operating within the juvenile justice system warrants attention ([95]). Practitioners in psychiatry, psychology, and law enforcement must collaborate with politicians and other stakeholders to develop and implement customised interventions for the effective screening and treatment of juveniles ([11]).

Opinions regarding treatment methods are, nonetheless, varied. According to research conducted by [43] ([43]), cognitive behavioural therapy, social skills training, and parent management training can effectively address symptoms of ADHD and ODD (obsessive–compulsive disorder). Conversely, another set of studies ([79]; [90]; [94]) suggest that a combination of medication and psychological interventions proves effective. They highlight that while pharmacological treatments are frequently employed for managing ADHD symptoms, relying solely on medication is unlikely to deter youth from engaging in RSB ([90]). Furthermore, ADHD medication can lower criminality rates among forensic individuals with ADHD, aiming to prevent and disrupt maladaptive developmental trajectories ([79]).

## 4. Discussion

Because it is common for children with ADHD to be impulsive or hyperactive while trying to control their behaviour, which may increase the likelihood of developing conduct problems, the role of sociodemographic factors was under-represented in the studies examined. Regarding gender, most studies focused on male perpetrators and did not examine only female perpetrators. The results of gender comparisons showed that males were over-represented in the less severe ADHD subtype, while females were over-represented in the more severe ADHD subtype. Female offenders showed more severe ADHD, often in combination with other behavioural problems. This may suggest that ADHD symptoms in women tend to be more severe or complex, possibly with multiple co-occurring problems such as other behavioural disorders. This difference may contribute to different diagnostic and treatment practices between the sexes, as more severe forms in women may receive delayed or different diagnoses ([87]; [102]). However, adverse childhood experiences showed no significant difference. The prevalence and severity of ADHD may differ by gender, with adverse childhood experiences often associated with delinquent behaviour, having similar effects on both sexes ([2]; [96]). Another study found no significant difference in the age of first offending in boys and girls. Children diagnosed with ADHD, regardless of gender, were more likely to commit crimes. This confirms that ADHD is a strong risk factor for criminal behaviour and that there is a clear association between ADHD and juvenile delinquency for both males and females. Young people with ADHD may be more prone to impulsive, uncontrollable behaviours that can lead to crime ([5]; [37]).

Sociodemographic factors and the role of the family have received attention in only a few studies. When examining the relationship between ADHD and violent behaviour, it can be seen that an ADHD diagnosis may be a significant contributor to chronic violent offending. Symptoms of ADHD, such as impulsivity, attention problems, and problems with emotion regulation, may contribute to increased violent behaviour, especially in the long term ([32]; [83]). The uncontrolled impulses and attention deficit often seen in ADHD can contribute to the repetitive nature of offending, particularly if the disorder is not adequately managed ([51]). Race and socioeconomic status (SES) did not significantly affect the frequency of offending or different levels of offending. An ADHD diagnosis is an independent and stronger risk factor for criminal behaviour than traditionally important social factors. This may also suggest that although sociodemographic factors such as poverty, family background, or race generally influence criminality, the presence of ADHD may have an independent and independently outweighing effect on the commission of crime ([75]; [81]).

It can, therefore, be concluded that offenders with ADHD start to commit crimes at an earlier age (approximately one and a half to two years earlier) than offenders in the general population. This is in line with the literature, which consistently shows that ADHD is associated with earlier onset of disruptive and delinquent behaviour and contributes to life-persistent delinquency rather than to transient delinquency during adolescence ([61]; [69]; [82]).

There are also inequalities in comorbidity, educational attainment, and living conditions among adolescents. Young people diagnosed with ADHD had lower educational attainment and family problems. Individuals with ADHD are often younger, have lower educational attainment, and face higher unemployment rates ([22]; [42]). They also had a younger age at offending compared to young people without ADHD. The lower educational attainment and family problems of young people diagnosed with ADHD suggest that ADHD is not only a neurobiological condition but that social and environmental factors also contribute to their difficulties ([94]; [104]). Symptoms of ADHD, such as impulsivity, attention problems, and organisational difficulties, can make school performance more difficult, leading to lower educational attainment later on ([24]). Family problems, such as dysfunctional family environments or low-income families, can exacerbate this situation by not providing a stable background for improving school performance ([17]; [55]).

Psychosocial factors, such as age at first conviction, play an important role in the persistence of violent behaviour, and early conviction increases the risk of violent crime. Early conviction can be associated with negative labelling for young people, deepening the distance between society and the individual ([13]). Labelling (stigma) can exacerbate criminal behaviour in the long term, as young people may internalise social rejection. If a young person enters the criminal justice system early, future rehabilitation, progression in the school system, or integration into the labour market may also be more difficult ([16]; [74]). Adverse childhood experiences also have a major impact on the behaviour of young offenders, as high levels of ACEs are present between the onset of intense offending and early offending. Children affected by ACEs often develop coping strategies, which may include aggressive patterns of behaviour to protect themselves or respond to threats ([32]; [83]). ACEs create traumatic experiences that often lead to later mental disorders (e.g., PTSD, anxiety, depression). These mental health problems are often associated with delinquent behaviour, as young people do not have adequate coping mechanisms to process their trauma ([31]; [48]).

Research shows that ADHD and its comorbidities have a significant impact on the behaviour and mental health of juvenile offenders. Several studies have shown that young people with ADHD often experience serious behavioural problems, including internalising (e.g., anxiety, depression) and externalising (e.g., aggression, conduct disorder) problems. Impulsivity and attention disorders may contribute to young people responding aggressively to situations ([92]). Increased irritability, difficulty with emotion regulation, and sudden outbursts of anger are often present in people with ADHD ([25]). Young people with ADHD are often characterised by behavioural problems such as misbehaviour, resistance to authority, problems at school, or difficulties in social relationships. These disorders can be a precursor to criminal behaviour ([36]).

ADHD symptoms are often associated with other psychiatric disorders, such as impulsivity and social withdrawal. Individuals with ADHD are particularly prone to anxiety, depression, and impulsive behaviour ([57]; [99]). ADHD is often associated with other mental disorders, such as depression, which is closely linked to juvenile delinquency. Young people with ADHD who have depression have lower self-esteem and more severe depressive symptoms ([66]). In addition to depression, OCD and other psychiatric disorders are also common among individuals with ADHD, such as CD (conduct disorder), which increases the risk of delinquency ([45]).

Among young offenders, behavioural problems such as oppositional defiant disorder are also common and have been strongly associated with ADHD. Research findings suggest that behavioural and social problems among juveniles with ADHD span a broad spectrum. Young people with ADHD are more likely to experience rejection from their peers, which has a long-term negative impact on their social relationships ([52]). Inattentive behaviour means that young people with ADHD do not always follow subtle cues in social interactions, which can lead to misunderstandings ([18]). Children and young people with ADHD often struggle to develop appropriate social skills. This includes regulating emotions, expressing empathy, and managing conflict constructively. Peers tend to avoid those who have difficulty fitting into social norms ([15]). Other disorders associated with ADHD, such as opposition defiant disorder (ODD) or conduct disorder (CD), exacerbate peer problems. Young people with these disorders often react more aggressively to conflict, which increases rejection ([44]).

Although there are ambivalent results on crime and recidivism, some research suggests that offenders diagnosed with ADHD reoffend more quickly and commit more crimes than those without the disorder. Individuals with ADHD may be more likely to be in a society where rule-breaking behaviour is accepted or encouraged ([103]). Individuals with impulsivity and adjustment problems may be more susceptible to the effects of such environments, which may maintain delinquent behaviour in the longer term ([20]). Additionally, individuals with ADHD do not always receive the psychological and psychiatric support they need in the criminal justice system. Without this, they find it more difficult to exit criminal behaviour and are more likely to reoffend ([1]; [40]).

Drug use is also a significant problem among young people with ADHD. The combination of ADHD and substance use disorder (SUD) is associated with a particularly high risk of delinquency. The co-occurrence of ADHD and SUD can contribute to increased delinquency, and such young people often have inadequate social relationships ([72]). Attention deficit, impulsivity, and emotional instability may appear to be reduced in the short term by substance use, but this can quickly lead to addiction, which further impairs emotional and social functioning ([72]).

Psychopathic personality traits also play a prominent role in the behaviour of offenders with ADHD. Psychopathic traits such as impulsivity, manipulation, and pathological lying are often present in young people with ADHD and other psychiatric disorders ([46]; [65]). Research by [28] ([28]) highlighted that antisocial behaviour appears early in young people with ADHD and that antisocial personality traits can exacerbate criminal tendencies. Young people with ADHD and psychopathic traits have difficulty forming deep, emotionally based relationships, which can lead to isolation and a hostile worldview. This contributes to the perpetuation of antisocial behaviour and a complete lack of empathy when harming others ([101]).

The majority of juvenile ‘delinquents’ are children with psychosocial needs, often living on the streets and not attending school. This phenomenon is a feature of the developing world, as in most HICs, those caught for status and other minor offences are often diverted towards reintegration through non-prison alternatives (diversion programmes) and juvenile correctional forms of incarceration are primarily reserved for young people charged or convicted of more serious property or personal offences.

It can, therefore, be concluded that, in the long run, ADHD is associated with a decline in educational aspirations, with far-reaching negative consequences for socioeconomic outcomes in adulthood, and may also be a strong predictor of a greater propensity to commit crime ([86]).

Overall, ADHD and its comorbidities have complex effects on adolescents’ behaviour and social inclusion. Behavioural problems associated with psychiatric disorders, such as aggression, anxiety, depression, and substance abuse, all contribute to increased risk of delinquency. Research emphasises that the treatment of ADHD should also take into account the presence of other comorbidities, as they can have a complex impact on young people’s lives.

From a methodological point of view, it should be noted that the results of self-completion questionnaires, diagnostic interviews, and structured scales have different validity and reliability, which may bias the results. In the case of self-completion questionnaires, the responses of individuals or caregivers may be influenced by memory biases, social conformity pressures, or the respondent’s own subjective interpretation. This can be particularly problematic for questions where we want to cluster patients along with behavioural problems. Self-completion questionnaires are not a substitute for structured interviews or clinical assessments, especially in diagnosing complex disorders such as ADHD or comorbidities. Clinical diagnostic measures should be used in research to increase reliability and validity. It should also be mentioned that in some studies, it was not clear what criteria were used to diagnose ADHD or other disorders, which reduces the validity of the data. Some studies have used strict diagnostic criteria, while others have used more flexible categorisation, which may distort prevalence data. Different sampling strategies (e.g., juvenile correctional, prison youth vs. school population) can significantly affect the prevalence and manifestation of ADHD and comorbid disorders. In addition, not all studies used a control group, making it difficult to generalise the results. In many cases, comorbid disorders (e.g., depression, PTSD, anxiety) have not been analysed in detail, which may bias the results and the understanding of the impact of ADHD. Socioeconomic factors, such as family background or school environment, have not received sufficient attention when interpreting the results. However, the consideration of these factors is of paramount importance, especially in the implementation of social reintegration and psychological support.

Although this review offered a detailed comparison between the studies included in the literature, the research is not without limitations. Limiting the review to English-language publications may have restricted the identification of relevant research. The exclusion of grey literature and non-English publications might have led to a publication bias, potentially omitting relevant data from under-represented regions or populations. Additionally, a substantial portion of the reviewed studies lacked control groups, limiting the ability to draw causal inferences. Furthermore, due to the variability and diversity of the included studies, it was not possible to calculate pooled sample sizes or effect sizes.

## 5. Conclusions

The above observations suggest that early identification and intervention are of paramount importance for young people with ADHD. Early diagnosis and appropriate treatment are essential to prevent delinquency. For young people with ADHD, there is a need for programmes that focus on addressing behavioural problems, developing social skills, and addressing mental health. Appropriate treatment and support programmes that target mental health, academic achievement, and social skills can help prevent later quality-of-life deterioration, unemployment, and delinquent behaviour. Strengthening the family environment and increasing support at school are also key to improving the situation. Psychological support (e.g., cognitive behavioural therapy, impulse control training, social skills development) and psychiatric treatments (e.g., medication for attention deficit disorder and comorbidities) can be effective in reducing the risk of crime.

Suggestions for solutions include the following:Targeted treatment and therapy: ADHD-specific psychotherapy programmes and medication can reduce impulsive behaviour.Skills development: Training in social and problem-solving skills can help improve social relationships and self-control.Structured environment: Building support systems that help people with ADHD live in a structured, predictable environment.Rehabilitation programmes: Specific programmes are needed in the justice system that consider ADHD-specific challenges.

## Figures and Tables

**Figure 1 behavsci-15-01044-f001:**
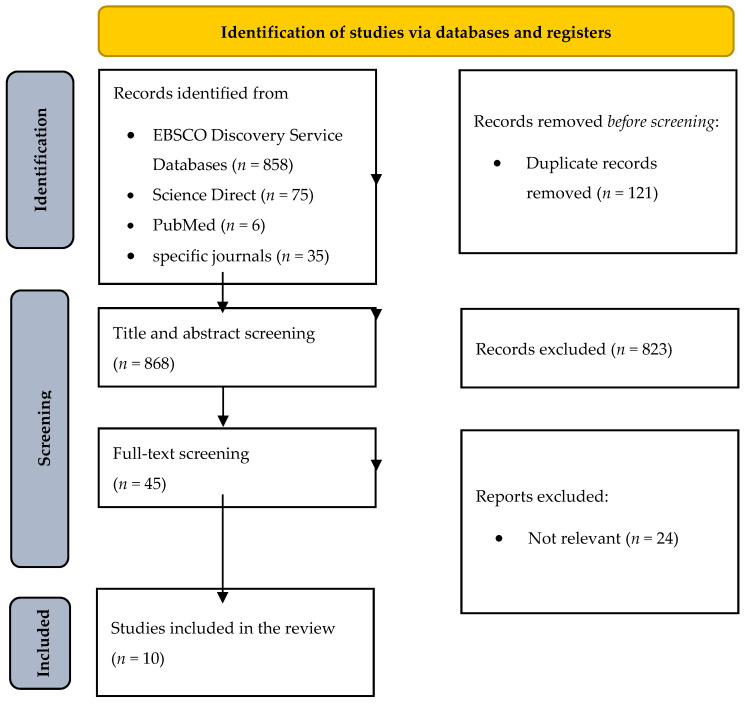
Preferred Reporting Items for Systematic Reviews and Meta-Analyses (PRISMA) diagram.

## Data Availability

Not applicable.

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
