# Peer review of "The Interplay Between Juvenile Delinquency and ADHD: A Systematic Review of Social, Psychological, and Educational Aspects"

_behavsci, 2025, doi:10.3390/bs15081044_

Round 1
Reviewer 1 Report
Comments and Suggestions for Authors
- Use "clear" or "convincing" rather than unequivocal, especially considering the rest of the paragraph discusses how sharing common covariates like ACEs may mean that the correlation between ADHD and delinquency may be illusory. (Page 2, Line 71)
- The literature uses criminological theory to explain how ADHD may be linked to delinquency. The presentation of these theories is seamless in its arguing how ADHD could develop as a risk factor for delinquency
- A major strength of this systematic review is that it followed the PRISMA guidelines
- Add "observational" with interview and survey as examples subset by parentheses (page 2, study design bullet point)[note: the pages in the version provided for review were not numbered consecutively; hence the reference to page 2 again]
- A detailed description of the search terms and databases used for identifying potential studies for the review is well written and comprehensive. Figure 1 provides a clear rendering of what happened to the pool of studies as the eligibility criteria were applied. One small equivocation relates to lack of detail in the records excluded box in Figure 1. N = 823 studies excluded raises concerns regarding reasons for exclusion and whether these may have biased the study’s findings.
- The paragraph in section 3.1 that discusses adverse childhood events is particularly striking. It is suggested that more detail be added to this paragraph.
- The last paragraph before section 3.3 ends with “which reflect significant underlying social issues”. This phrase is ambiguous and instead of adding additional text to clarify it, it is suggested that this phrase be eliminated from the text.
- In section 3.4, the following sentence need revision to improve clarity: “In the case of some studies, the diagnostic criteria used were un-clear, which determined the criteria used to assign offenders to ADHD in the study”
- There are a lot of acronyms used throughout the paper (e.g., CD, ASD, RD). These are difficult to remember so it is suggested that the paper remind the reader what these diagnoses are. A good place to do this would be in the Academic Achievement section of the results.
- In section 3.7, the following appears: “Some studies have linked it to a diagnostic criteria system” The antecedent to “it” is unclear so replace the pronoun with the noun it represents to improve clarity of writing.
- In section 3.7, “Suggestions” appears by itself on its own line. What suggestions are being made? Are they the several paragraphs that follow? If so, Write a transitional sentence that explains what the suggestions are. -or-
- The last 4 paragraphs of 3.7 do not appear to be based in the narrative that led the paragraph. Moving and integrating them into Section 4. Discussion would seem a more appropriate place for them.
- In the reference list it is customary to place an * at the beginning of the references that were summarized in the review article.
Author Response
Dear Reviewer 1,
thank you very much for the review comments. Based on the notifications, the following modifications has been carried out:
Reviewer 1: Use "clear" or "convincing" rather than unequivocal, especially considering the rest of the paragraph discusses how sharing common covariates like ACEs may mean that the correlation between ADHD and delinquency may be illusory. (Page 2, Line 71)
Authors: Thank you very much for your kind comment. We modified the word “unequivocal” to “clear”.
Reviewer 1: The literature uses criminological theory to explain how ADHD may be linked to delinquency. The presentation of these theories is seamless in its arguing how ADHD could develop as a risk factor for delinquency
Authors: Thank you very much for your suggestion. We have expanded this paragraph to explore the links between ADHD and higher levels of offending among juveniles
Reviewer 1: A major strength of this systematic review is that it followed the PRISMA guidelines
Authors: Thank you very much for your kind feedback.
Reviewer 1: Add "observational" with interview and survey as examples subset by parentheses (page 2, study design bullet point)[note: the pages in the version provided for review were not numbered consecutively; hence the reference to page 2 again]
Authors: Thank you very much for your kind suggestion, we carried out this modification.
Reviewer 1: A detailed description of the search terms and databases used for identifying potential studies for the review is well written and comprehensive. Figure 1 provides a clear rendering of what happened to the pool of studies as the eligibility criteria were applied. One small equivocation relates to lack of detail in the records excluded box in Figure 1. N = 823 studies excluded raises concerns regarding reasons for exclusion and whether these may have biased the study’s findings.
Authors: Thank you very much for your notification. In the 3. Results section, we state as follows: During the title and abstract screening phase, the majority of the 823 excluded records were omitted because they (a) did not focus on juvenile offenders, (b) did not address ADHD explicitly, (c) were review articles or grey literature, or (d) lacked empirical data relevant to our outcomes of interest. Besides, during the search, we asked the search engine (EBSCO) to search the keywords with “att text” option rather than “abstract” which may have led that we had more records that had irrelevant focus.
Reviewer 1: The paragraph in section 3.1 that discusses adverse childhood events is particularly striking. It is suggested that more detail be added to this paragraph.
Authors: Thank you very much for your suggestion. We have expanded this paragraph concerning ACE.
Reviewer 1: The last paragraph before section 3.3 ends with “which reflect significant underlying social issues”. This phrase is ambiguous and instead of adding additional text to clarify it, it is suggested that this phrase be eliminated from the text.
Authors: Thank you very much for your suggestion. We deleted this phrase.
Reviewer 1: In section 3.4, the following sentence need revision to improve clarity: “In the case of some studies, the diagnostic criteria used were un-clear, which determined the criteria used to assign offenders to ADHD in the study”
Authors: Thank you very much for your kind suggestion, we carried out this modification.
Reviewer 1: There are a lot of acronyms used throughout the paper (e.g., CD, ASD, RD). These are difficult to remember so it is suggested that the paper remind the reader what these diagnoses are. A good place to do this would be in the Academic Achievement section of the results.
Authors: Thank you very much for your kind suggestion, we reminded the readers in this section.
Reviewer 1: In section 3.7, the following appears: “Some studies have linked it to a diagnostic criteria system” The antecedent to “it” is unclear so replace the pronoun with the noun it represents to improve clarity of writing.
Authors: Thank you very much for your kind notification. We modified this phrase to “Some studies used a diagnostic criteria system”.
Reviewer 1: In section 3.7, “Suggestions” appears by itself on its own line. What suggestions are being made? Are they the several paragraphs that follow? If so, Write a transitional sentence that explains what the suggestions are. -or- The last 4 paragraphs of 3.7 do not appear to be based in the narrative that led the paragraph. Moving and integrating them into Section 4. Discussion would seem a more appropriate place for them.
Authors: Thank you very much for your kind notification. We deleted “Suggestion” there since it was not on the right place.
Reviewer 1: In the reference list it is customary to place an * at the beginning of the references that were summarized in the review article.
Authors: Thank you very much for your kind suggestion. We marked the references concerned with an asterisk.
We hope our modifications are correct and we could provide a high-quality paper which can be considered for publication.
Kind regards,
Karolina Eszter Kovács (senior lecturer, University of Debrecen)
Reviewer 2 Report
Comments and Suggestions for Authors
This manuscript was very well-written and with only a few minor editorial adjustments I feel it will be suitable for publication.
The writing style was clear and easy to follow. The rationale for the need for the study was clear – the interplay between ADHD and juvenile delinquency is complex and overlays with familial, psychological, social and behavioral are certainly nuanced.
The English writing style was free of jargon and grammatical errors. I did find a few instances however of switching between tenses ("were used", "should have been written") and between passive and active voice. APA style was also closely followed.
The methodology provided a strong foundation for a systematic review and follows several best-practice guidelines. Specifically, following PRISMA and referencing Moher et al., (2015) provided transparency and methodological credibility. The included flowchart (Figure 1) demonstrated adherence to systematic reporting standards. Additionally, the use of the PICOS framework (Population, Intervention, Comparison, Outcomes, Study Design) enhances clarity and systematic decision-making. Non-research sources (e.g., commentaries, editorials) are rightly excluded. However, the methodology lacks information on the publication date range for included studies. Were all years eligible or was there a cutoff (e.g., studies from the last 20 years)?
Only three databases were searched. Key sources like PsycINFO, ERIC, or Criminal Justice Abstracts were not included—possibly limiting scope in psychology, education, and criminology. Similarly, the reference to “top 30 journals” is vague; the description should indicate which disciplines were searched, which percentile criteria were used, and were these journals hand-searched.
Regarding the search string logic, the keywords used were stated, but Boolean logic and truncation details are not reported (e.g., was it juvenile delinquen* AND ADHD?; offend*, etc.). This lack of specificity limits replicability.
Although only English-language studies were included, this criterion should have been applied in the initial search or clearly justified to avoid language bias. Similarly, there is no mention of grey literature searching. Excluding dissertations, conference proceedings, or government reports may limit comprehensiveness, especially in a niche area like ADHD in juvenile justice.
The references should be double-checked for accuracy and errors; as one example, the Sampson reference is included as SAMPSON, and the title includes capital letters throughout.
Overall, these are relatively minor suggestions for revision to an already well-written manuscript.
Author Response
Dear Reviewer 2,
thank you very much for the review comments. Based on the notifications, the following modifications has been carried out:
Reviewer 2: This manuscript was very well-written and with only a few minor editorial adjustments I feel it will be suitable for publication.
Authors: Thank you very much for your kind feedback.
Reviewer 2: The writing style was clear and easy to follow. The rationale for the need for the study was clear – the interplay between ADHD and juvenile delinquency is complex and overlays with familial, psychological, social and behavioral are certainly nuanced.
Authors: Thank you very much for your kind evaluation.
Reviewer 2: The English writing style was free of jargon and grammatical errors. I did find a few instances however of switching between tenses ("were used", "should have been written") and between passive and active voice. APA style was also closely followed.
Authors: Thank you very much for your positive reinforcement.
Reviewer 2: The methodology provided a strong foundation for a systematic review and follows several best-practice guidelines. Specifically, following PRISMA and referencing Moher et al., (2015) provided transparency and methodological credibility. The included flowchart (Figure 1) demonstrated adherence to systematic reporting standards. Additionally, the use of the PICOS framework (Population, Intervention, Comparison, Outcomes, Study Design) enhances clarity and systematic decision-making. Non-research sources (e.g., commentaries, editorials) are rightly excluded. However, the methodology lacks information on the publication date range for included studies. Were all years eligible or was there a cutoff (e.g., studies from the last 20 years)?
Authors: Thank you very much for your, we have added the information, the date range was 2004-2024.
Reviewer 2: Only three databases were searched. Key sources like PsycINFO, ERIC, or Criminal Justice Abstracts were not included—possibly limiting scope in psychology, education, and criminology. Similarly, the reference to “top 30 journals” is vague; the description should indicate which disciplines were searched, which percentile criteria were used, and were these journals hand-searched.
Authors: Thank you very much for your kind notification. We used rater EBSCO Discovery service as it provides access to papers published in journals that are indexed in various databases (85 databases). This includes ERIC too, therefore, we considered ERIC as a database screened. Databases that are not freely available were not used in the study for financial reasons. Therefore, PsycINFO and Criminal Justice Abstracts have been excluded from the search. Regarding the choice of the top 30 journals: we searched for relevant journals in the field that are indexed in Scopus and Web of Science, and based on their metrics, we chose the 30 most relevant to specifically search in those journals too. We added a table to the appendix with the quality of these journals.
Reviewer 2: Regarding the search string logic, the keywords used were stated, but Boolean logic and truncation details are not reported (e.g., was it juvenile delinquen* AND ADHD?; offend*, etc.). This lack of specificity limits replicability.
Authors: Thank you very much for your kind suggestion. We modified the relevant part (2.1. Literature review).
Reviewer 2: Although only English-language studies were included, this criterion should have been applied in the initial search or clearly justified to avoid language bias. Similarly, there is no mention of grey literature searching. Excluding dissertations, conference proceedings, or government reports may limit comprehensiveness, especially in a niche area like ADHD in juvenile justice.
Authors: Thank you very much for your kind suggestion. We added a justification to these parts in 2.2. Inclusion and exclusion criteria, but we also mention this as a limitation in the Discussion.
Reviewer 2: The references should be double-checked for accuracy and errors; as one example, the Sampson reference is included as SAMPSON, and the title includes capital letters throughout.
Authors: Thank you very much for your kind suggestion. We double-checked the references.
Reviewer 2: Overall, these are relatively minor suggestions for revision to an already well-written manuscript.
Authors: Thank you very much for your valuable thoughts and positive feedback.
We hope our modifications are correct and we could provide a high-quality paper which can be considered for publication.
Kind regards,
Karolina Eszter Kovács (senior lecturer, University of Debrecen)